# Comparative study on abortion characteristics of *Nsa* CMS and *Pol* CMS and analysis of long non-coding RNAs related to pollen abortion in *Brassica napus*

**Man Xing**[1,2], **Zechuan Peng**[1,2], **Chunyun Guan**[1,3], **Mei Guan**[1,2,3]*

**1** Hunan Branch of National Oilseed Crops Improvement Center, Changsha, China, **2** College of Agriculture, Hunan Agricultural University, Changsha, China, **3** Southern Regional Collaborative Innovation Center for Grain and Oil Crops in China, Changsha, China

* gm7142005@hunau.edu.cn

**Data Availability Statement:** All relevant data are within the paper and its Supporting Information files.

## Abstract

Cytoplasmic male sterile system (CMS) is one of the important methods for the utilization of heterosis in *Brassica napus*. The involvement of long non-coding RNAs (lncRNAs) in anther and pollen development in *B.napus* has been recognized, but there is little data on the involvement of lncRNAs in pollen abortion in different types of rapeseed CMS. The present study compared the cytological, physiological and biochemical characteristics of *Nsa* CMS (1258A) and *Pol* CMS (P5A) during pollen abortion, and high-throughput sequencing of flower buds of different sizes before and after pollen abortion. The results showed that insufficient energy supply was an important physiological basis for 1258A and P5A pollen abortion, and 1258A had excessive ROS (reactive oxygen species) accumulation in the stage of pollen abortion. Functional analysis showed that Starch and sucrose metabolism and Sulfur metabolism were significantly enriched before and after pollen abortion in 1258A and P5A, and a large number of genes were down-regulated. In 1258A, 227 lncRNAs had *cis*-targeting regulation, and 240 *cis*-target genes of the lncRNAs were identified. In P5A, 116 lncRNAs had *cis*-targeting regulation, and 101 *cis*-target genes of the lncRNAs were identified. There were five lncRNAs *cis*-target genes in 1258A and P5A during pollen abortion, and *LOC106445716* encodes β-D-glucopyranosyl abscisate β-glucosidase and could regulate pollen abortion. Taken together, this study, provides a new perspective for lncRNAs to participate in the regulation of *Nsa* CMS and *Pol* CMS pollen abortion.

## Introduction

Rapeseed (*B.napus*) is an important oil crop. Its seeds are not only rich in oil and protein but can also be used as a source of protein feed and are widely planted all over the world. The yield can be substantially increased using crop heterosis. Cytoplasmic male sterility (CMS) is one of the most important methods to utilize heterosis. The types of CMS in *B. napus* are *pol* CMS [1], *Shaan2A* CMS [2], *nap* CMS [3], *Ogu* CMS [4], *hau* CMS [5], *NCa* CMS [6], *SaNa-1A*

**Funding:** This work was supported by the China Agriculture Research System of MOF and MARA (CARS-13), the High-Tech Research and Development Program 863 (grant number 2011AA10A104).The funders had no role in study design, data collection and analysis, decision to publish, or preparation of the manuscript.

**Competing interests:** The authors have declared that no competing interests exist.

CMS [7] and *Nsa* CMS [8]. During the preparation of rapeseed three-line hybrids, *Pol* CMS, *Shaan2A* CMS and *Ogu* CMS [9, 10] are the most widely used CMS systems. The molecular mechanism of pollen abortion in rapeseed *Nsa* CMS and *Pol* CMS systems has been widely studied, but what are the characteristics of lncRNA in rapeseed *Nsa* CMS and *Pol* CMS systems and how they play a role in the process of pollen abortion have not been reported.

LncRNA is a non-coding RNA (ncRNA) that > 200bp long, which is located in the nucleus or cytoplasm, and can play a role in the nucleus and cytoplasm to regulate gene expression [11]. LncRNAs typically do not participate in protein synthesis and can be divided into three categories according to their position in the genome relative to protein-coding genes: long intergenic ncRNAs (which do not overlap with protein-coding genes), long intron ncRNAs (synthesized from intron regions) and natural antisense transcripts (NAT, synthesized from the opposite chain of related genes) [12]. In the nucleus, lncRNAs regulate gene expression by affecting chromatin remodeling, epigenetic modification and alternative splicing, which can affect genome stability and nuclear domain organization [13–15]. LncRNAs can be divided into five types: antisense long chain non-coding RNA (antisense lncRNA), intron non-coding RNA (intronic lncRNA), intergenic long chain non-coding RNA (intergenic lncRNA), promoter-associated long chain non-coding RNA (promoter-associated lncRNA) and untranslated region (UTR)-associated RNA (UTR-associated lncRNA) [16]. LncRNAs in plants are similar to mRNAs. Most need to undergo 5' end capping and 3' polyadenosine acidification (5' capped and 3' polyadenylated). PolII-dependent lncRNA maturation requires capping, splicing and the addition of Poly-a tails, while PolIV/V-dependent lncRNA maturation is not needed and can be stabilized from primary PolII transcripts through alternative mechanisms [17–20]. The transcriptional regulation of lncRNAs is primarily regulated by RNA PolII. In addition, PolIII and plant-specific RNA PolIV/V can also be transcribed [19, 21, 22].

Previous studies often underestimated the function of lncRNAs, and compared with mRNAs or miRNAs, the level of expression of lncRNAs was lower. With more study, their biological functions and related molecular mechanisms began to appear in plants. With the in-depth study of plant ncRNAs, many lncRNAs have been identified in plants, and some lncRNAs are considered to be related to plant fertility. Researchers identified a long-day specific male fertility-related RNA (LDMAR) (long-day-specific male-fertility-associated) in rice (*Oryza sativa*), which is a 1236-base lncRNA that can regulate photoperiod-sensitive male sterility (PSMS) in rice [23]. A total of 865 differentially expressed lncRNAs (DE lncRNAs) were identified during the development of cotton (*Gossypium hirsutum*) anthers that correspond to 1,172 *cis*-target genes and are involved in pollen development [24]. Some lncrnas are thought to be involved in the regulation of pollen development in cytoplasmic male sterile plants. Based on the construction of a lncRNAs regulatory network in cotton CMS and restorer lines, the abnormal expression of lncRNAs and their target genes was found to lead to pollen abortion, and a miRNA-lncRNA-mRNA network could be involved in CMS and fertility restoration [24, 25]. In addition, lncRNAs for reproductive development were identified in strawberry (*Fragaria* x *ananassa*) [26], tomato (*Solanum lycopersicum*) [27], maize (*Zea mays*) [28], trifoliate orange (*Poncirus trifoliata*) [29], spinach (*Spinacea oleracea*) [30], rapeseed [31] and other plants. They not only participate in pollen development but also play an important role in many fertility-related processes, such as flowering and flower bud development.

In recent years, lncRNAs have been proven to regulate the pollen development of rapeseed. In rapeseed, 14 lncRNAs were highly co-expressed with 10 pollen-related coding genes with known functions. Two lncRNAs were identified as functional targeting mimics of miR160 and play a role in pollen development [31]. The study of CMS transcripts among different types of rapeseed is very important for the identification of lncRNAs and their gene regulatory network involved in anther or pollen abortion. *Pol* CMS is owing to the undifferentiation of sporogenic

cells, absence of pollen sac structure and pollen production during the early stage of anther development [32]. Candidate genes and lncRNAs related to cell differentiation were found in *Pol* CMS. The abnormal expression of these genes and lncRNAs may result in the absence of spore cells [33]. Recently, studies on the male sterile system of *B. napus* revealed new insights into the mechanism of non-coding RNAs (ncRNAs) involved in the development of anther and pollen. Through the comparative study of the genic male-sterile lines *Pol* CMS and *Ogu* CMS of Chinese cabbage (*B. rapa* subsp. *pekinensis*), 361 DE lncRNAs were identified. Specific regulatory networks related to anther cell differentiation, meiosis and cytokinesis, pollen wall formation and tapetum development were constructed [33]. Further studies confirmed that the target gene of lncRNAs can act on rapeseed pollen formation, and overexpression in the rapeseed pollen tube can reduce the growth of pollen tubes, leading to the shortening of pollen tubes, and finally affecting the fruiting of rapeseed pollen tubes [31, 33].

*Nsa* CMS can be divided into two types according to its origin. One was obtained by somatic hybridization between "Zhongshuang4" (*B. napus*) and "Yeyou18" (*Sinapis arvensis*). And the other derived from a cross between "Xinjiang Yeyou"(*S. arvensis*) and "Xiangyou15" (*B. napus*) [34, 35]. The results of cytological observation showed that *Nsa* CMS uninucleate pollen grains all aborted and could not develop into mature pollen grains [35–37]. *Pol* CMS is owing to abnormal differentiation of the sporogenic cells during the early stage of anther development, which cannot further differentiate into inner walls, middle layers and tapeta, and no pollen sacs form [32]. Abnormal pollen development at any stage from the differentiation of sporogonia to the formation of mature pollen grains will lead to pollen abortion. *Pol* CMS in the early stage of pollen development and *Nsa* CMS in the late stage of pollen development are suitable materials to study pollen abortion in rapeseed.

In order to further explain the pollen abortion mechanism of two different CMS systems (*Pol* CMS and *Nsa* CMS) in rapeseed, we observed in detail the process of pollen abortion. Sucrose, soluble sugar, ATP (adenosine triphosphate), Pro (proline), MDA (malondialdehyde), POD (peroxidase), SOD (superoxide dismutase) and $H_2O_2$ were determined in different stages of pollen abortion, and the differences among different abortion types were compared. According to the results of comparative study, flower buds before and after pollen abortion were selected for RNA transcriptome sequencing, and the molecular mechanisms of lncRNAs and mRNAs related to pollen development in *B. napus* were analyzed in detail. The differentially expressed heterologous lncRNAs and its target genes related to pollen development were identified in *Pol* CMS and *Nsa* CMS. Our results deeply analyzed the mechanism of pollen abortion of *Pol* CMS and *Nsa* CMS from the aspects of cytology, physiology, biochemistry and lncRNAs, and explained the specific interaction of lncRNAs and mRNAs related to pollen development, which can broaden the understanding of the molecular mechanism of CMS in *B. napus*.

## Methods

### Plant materials

The cytoplasmic male sterile line *Pol* CMS (P5A) and maintainer line (P5B) and restorer line P26R of *B. napus* were provided by Tingdong Fu of Huazhong Agricultural University (Wuhan, China). *Nsa* CMS (1258A) and maintainer line (1258B) were provided by the Oil Crop Research Institute of Hunan Agricultural University (Changsha, China). 1258A was a male sterile plant with *Nsa* cytoplasm obtained by crossing Xinjiang Yeyou (*S. arvensis*) as the female parent and Xiangyou 15 (*Brassica napus* L.) as the male parent, the sterile cytoplasm was maintained by backcross. Finally, a pair of Isonuclear heterologous cytoplasmic male sterile lines of rapeseed were selected. The three varieties were planted in the experimental field of Hunan Agricultural University according to the plot, and the field management was

conducted according to the local production conditions. The flower buds at different stages of pollen development were sampled at the flowering stage of rapeseed. Small flower buds (SB, ≤ 2mm) corresponded to the pre-pollen mother cell stage of pollen development, and medium-sized flower buds (MB, 4mm) corresponded to the mature pollen grain stage of pollen development. Buds were collected from three different plants for transcriptomic profiling. Each bud sample was collected from the same plant, with three repetitions. The collected bud samples were immediately frozen in liquid nitrogen and stored at -80 .

## Observation of floral organ and anther development

During the flowering stage of rapeseed, the blooming flowers and buds of 1258A, 1258B, P5A and P5B were taken, and the petals were peeled off with tweezers. The morphological differences of flower organs and stamens between male sterile line and maintainer line were observed by stereoscopic microscope (Motic, K4000, Hong Kong, China). The anthers of different sizes of 1258A and P5A were made into semi-thin sections to observe the process of pollen abortion. The preparation and staining methods of semi-thin sections were carried out according to the experimental method of Xing et al. [35]. The semi-thin sections of pollen at different developmental stages were observed by optical microscope (Olympus CX51, Olympus Corporation, Tokyo, Japan).

## Determination of physiological indexes in anthers

The physiological and biochemical indexes measured in anthers were sucrose, soluble sugar, ATP, Pro, MDA, POD, $H_2O_2$ and SOD. We took small flower buds (SB, tetrad stage, early stage of sterilization) and medium size flower buds (MB, mature pollen stage, late stage of sterilization) from sterile lines and maintainers, and measured physiological and biochemical indexes in anthers. The specific operation method is as follows: peeling off the bud with aseptic tweezers, quickly clipping the complete anther 0.1g, putting it into liquid nitrogen quick-freezing, and then storing it at -80. All our test kits are purchased from Sangon Biotech Co. (Shanghai, China), and the samples are processed according to the kit instructions (sucrose: D799789, soluble sugar: D799391, ATP: D799643, Pro: D799575, MDA: D799761, POD: D799591, $H_2O_2$: D799773, SOD: D799593), with three times of technical repetition.

## Transcriptome sequencing

High-throughput sequencing was performed on the buds of *Nsa* CMS (1258A), *Pol* CMS (P5A), and restorer lines (P26R) SB and MB stages, each with three biological replicates. The total RNA of rapeseed flower buds was extracted using TRIzol (Invitrogen, Carlsbad, CA, USA) according to the product instructions. An NEBNext® Ultra™ RNA Library Prep Kit for Illumina® (NEB, Ipswich, MA, USA) was used to construct mRNA-Seq and lncRNA-Seq libraries. Total RNA sample requirements and RNA-seq library construction methods were carried out according to the experimental method of Xing et al. [35]. RNA-Seq was performed on an Illumina Hi-Seq platform by Beijing BoaoJingdian Biotechnology Co., Ltd. (Beijing, China).

## Transcriptome assembly and lncRNA identification

Fastp [38] was used to filter the original data, delete the reads that did not meet the criteria for analysis criteria,including adapter pollution, low-quality bases and undetermined bases, and obtain the clean reads that can be used for subsequent comparison and analysis. Clean reads were compared with the rape reference genome sequence (Bra_napus_v2.0) by HISAT2 [38] analytical software. According to the results of reads alignment, StringTie [39] was used to

assemble the genes and transcripts. The expression read count was calculated by StringTie. The FPKM of genes and transcripts was then calculated. During this process, we simply filtered the expression as follows: (1) by removing the gene or transcript whose length < 100bp; (2) if the FPKM of the gene or transcript < 0.0001, it was regarded as 0. The genes expressed as 0 in all the samples were removed.

We took the following steps to screen and identify the lncRNAs: (1) Transcript length filtering, according to the threshold of lncRNAs length, potential assembly transcripts whose lengths< 200bp were filtered out. (2) FPKM value filtering: for the assembled transcript with only one exon, transcripts with FPKM ≥ 2 were kept. For the assembled transcript with multiple exons, the transcripts with FPKM ≥ 0.5 were kept. When the number of samples > 3, at least two samples were required to meet the screening criteria described above. The unknown transcript assembled by StringTie software was used for further coding capability filtering, and CPC, CNCI and HMMER software were selected to predict the coding ability. The transcripts with no coding ability predicted by the three software were novel lncRNAs [40, 41].

## Differential mRNA and lncRNA expression analyses

FPKM is the current standard method to measure gene expression in RNA-Seq. FPKM was calculated by StringTie software to detect the level of expression of all the transcripts, including mRNAs and lncRNAs [42]. Differentially expressed genes (DEGs) were analyzed by DEGseq software [43]. The screening criteria for DEGs/differentially expressed transcripts were: | log2FC| ≥ 1 and p≤ 0. 05. The average level of expression of at least one group in the case/control was ≥ 0. 5. The number of expressed samples accounts for two-thirds of the total number of samples, or at least one group of case/control had samples >two-thirds. In this case, the average level of expression of this group > 1. The mRNA and DEG underwent GO enrichment and KEGG pathway analyses.

## Target gene prediction and functional enrichment analysis of differential lncRNAs

We predicted the *cis*-target genes of differentially expressed lncRNAs. The software lncTar was used to predict whether there was targeted regulation on the basis of the lncRNA-mRNA relationship pairs obtained by co-expression analysis. LncRNAs may play a *cis* role in adjacent target genes [44]. We predicted the *cis*-target genes of lncRNAs. The *cis*-target gene of lncRNAs was enriched and analyzed by GO annotation and KEGG pathway analysis.

## Real-time quantitative reverse transcription PCR (qRT-PCR) validation

Total RNA was extracted from the flower buds at different stages of pollen development using a Plant Total RNA Isolation Kit (SangonBiotech, Shanghai, China) and reverse transcribed with TransScript®One-Step gDNA Removal and cDNA Synthesis SuperMix (TransGen, Beijing, China). The relative levels of expression of lncRNAs and mRNAs were verified by qRT PCR according to the instructions for the use of TransStart®Green qPCR SuperMix (Trans-Gen). Determination of related gene expression in a CFX96 Real-Time System (Bio-Rad, Hercules, CA, USA), and *Brassica napus Actin2.1* was used as an internal reference gene. The reactions were conducted at following conditions: 94˚C for 30 s, followed by 40 cycles of 94˚C for 5 s and 60˚C for 30 s. All reactions were performed with three replicates. We used Premier 6. 0 software to design primers and calculated the relative level of expression as described by Livak et al. [45]. The qRT PCR primers and results used are listed in S2 Fig in S1 File.

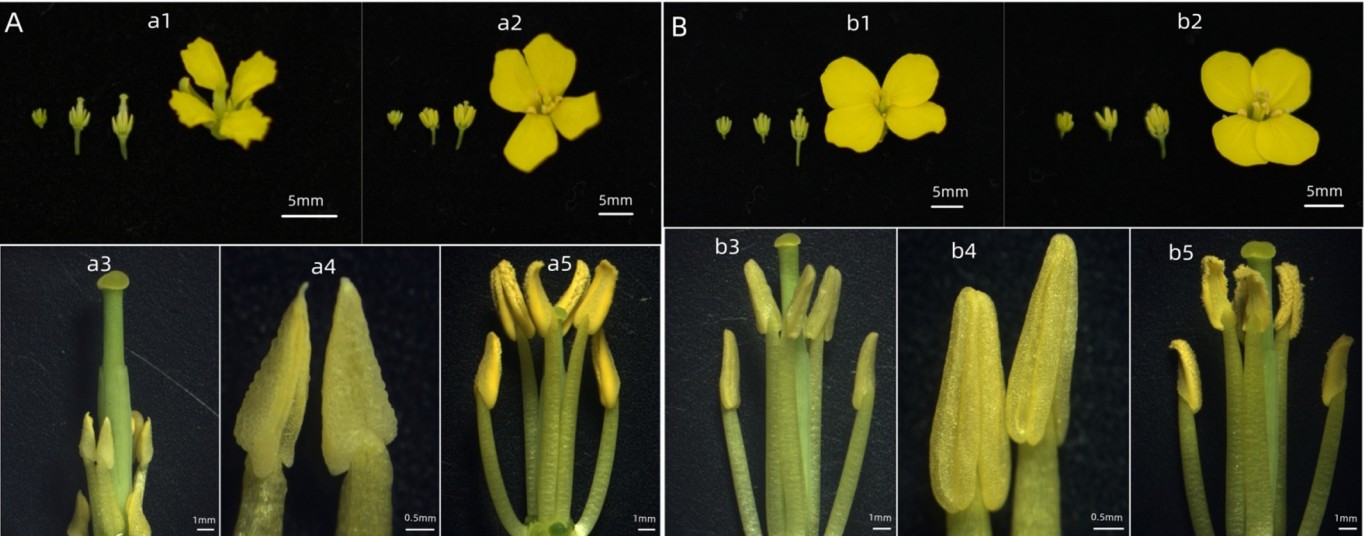

**Fig 1. Morphological observation of flower organs.** (A)The phenotype comparision between *Pol* CMS and maintainer. a1, a3, a4: Flowers of P5A. a2, a5: Flowers of P5B; (B) The phenotype comparision between *Nsa* CMS and maintainer. b1, b3, b4: Flowers of 1258A; b2, b5: Flowers of 1258B.

## Results

### Morphological and cytological characteristics of Nsa CMS and Pol CMS

The petals and stamens of *Pol* CMS P5A and P5B are different in size (Fig 1A). Compared with P5B, the petals of P5A are wrinkled, smaller and can not be completely extended. The stamens of P5A were significantly smaller than those of P5B, the anthers were triangular and no pollen was produced. *Nsa* CMS 1258A and maintainer line 1258B had the same flower organ size, and the size of petals and stamens was the same as that of maintainer line (Fig 1B). There was only pollen in 1258B anther, but no pollen in 1258A.

The cytological observation of 1258A and P5A pollen abortion showed that 1258A pollen abortion occurred in the uninucleate pollen grain stage (Fig 2). However, P5A pollen abortion occurred earlier. P5A pollen abortion is due to the failure of sporogenic cell differentiation, no differentiation into sporogenic cells, intermediate layer, endothelial layer and tapetum, no pollen sac structure. 1258A pollen abortion is reflected in the abnormal tapetum, mononucleate pollen grains can not form mature pollen grains. Although the pollen abortion periods of 1258A and P5A are different, they both belong to the type of pollen abortion, and the stamens have anthers and filaments.

### Physiological and biochemical characteristics of Nsa CMS and Pol CMS

The anthers of 1258A, 1258B, P5A and P5B (SB, ≤ 2mm, early stage of pollen abortion) and medium-size buds (MB, 4mm, late stage of pollen abortion) were taken, and the contents of physiological and biochemical indexes in anthers were determined (Fig 3). At the SB stage before pollen abortion, the sucrose content in the anther of maintainer line was significantly higher than that of male sterile line. At the MB stage after pollen abortion, the sucrose content in anther of P5B was significantly lower than that of P5A, but there was no significant change in 1258A and 1258B. The total sucrose content of anthers in the two maintainer lines decreased significantly from SB stage to MB stage, but there was no significant decrease in male sterile lines (Fig 3A). The soluble sugar content in P5A and P5B anthers decreased significantly from SB stage to MB stage, and the content in male sterile line was higher than that in

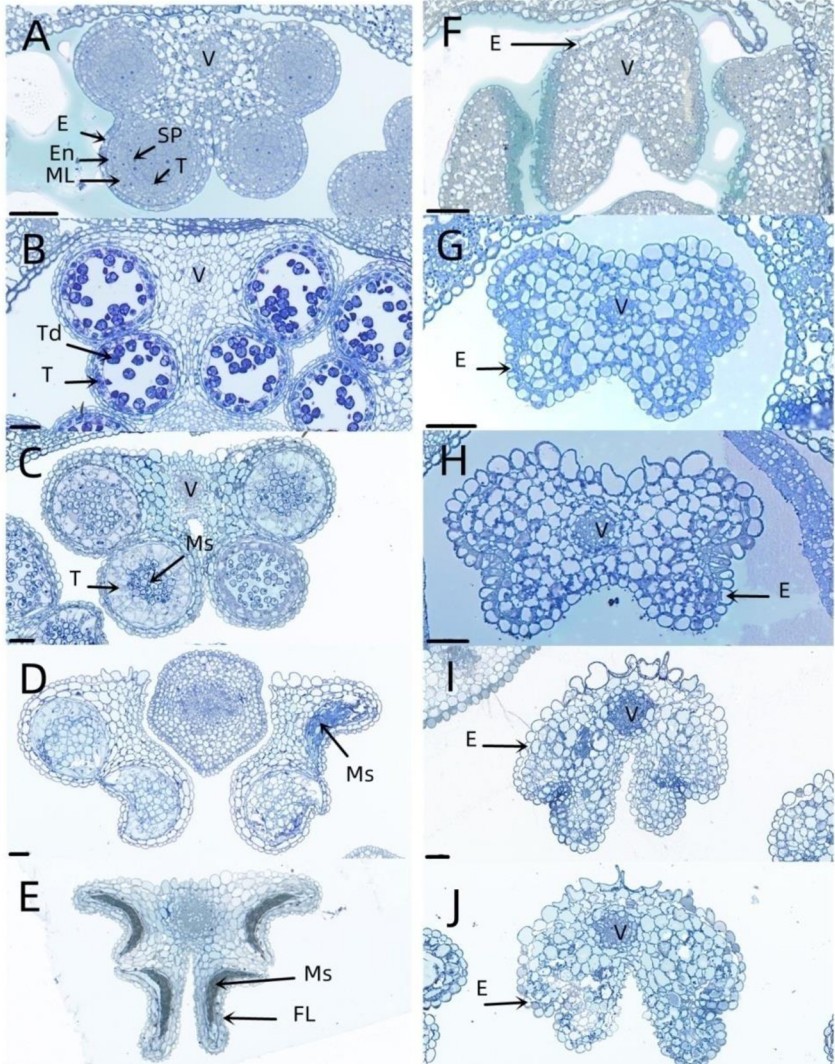

**Fig 2.** Semi-thin sections to observe anter of 1258A (A–E) and P5A (F–J) during pollen develop. (A, F) Sporogenous cell differentiation stage; (B, G) microspore tetrad stage; (C, H) single nucleus pollen grain stage; (D, I) mature pollen grain stage; (E, J) pollen release period. E, epidermis; En, endothecium; FL, fibrous layer; ML, middle layer; SP, sporogenous cell; T, tapetum; Td, Tetrad; Ms, microspore; V, vascular region. Bar = 50μm.

maintainer line. The soluble sugar content in 1258A anther decreased significantly from SB stage to MB stage, but there was no significant change in maintainer line (Fig 3B). Before pollen abortion period of 1258A and P5A, the content of ATP in anther was significantly lower than that of maintainer line (Fig 3C). It can be seen that male sterile lines are faced with insufficient supply of ATP in the process of pollen abortion. The Pro content of P5B and 1258B was significantly higher than that of male sterile lines before and after pollen abortion, P5B was significantly higher than 1258B. After pollen abortion, the content of Pro in P5A in MB stage was significantly higher than that in SB stage (Fig 3D). The MDA content in 1258A anther was significantly higher than that in maintainer line at SB stage, but significantly lower than that in maintainer line at MB stage after pollen abortion. The content of MDA decreased significantly

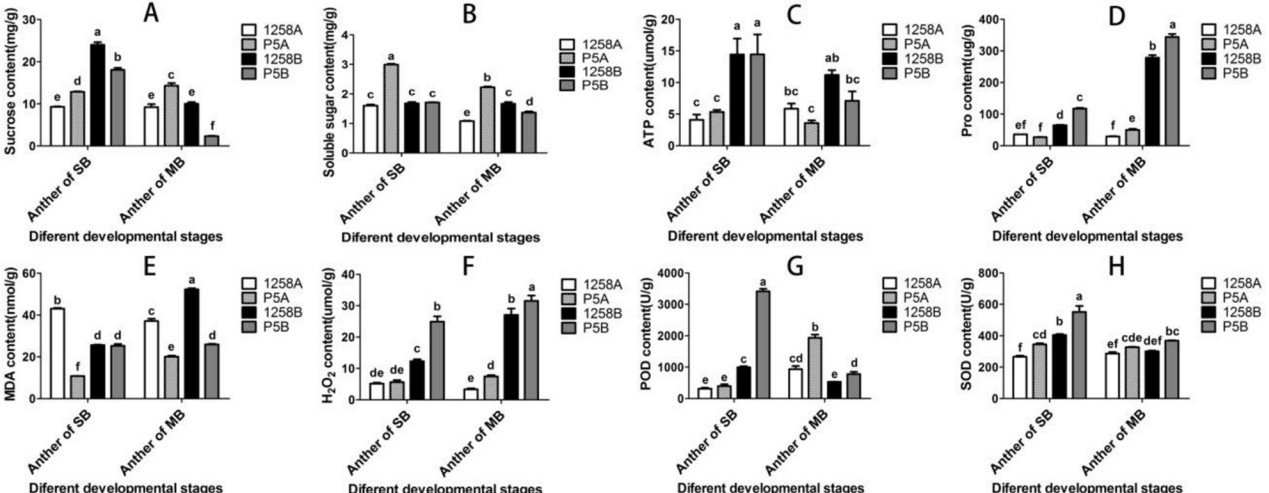

**Fig 3. Physiological and biochemical indexes for 1258A and P5A anthers at different periods of development.** A-F: sucrose, soluble sugar, ATP, Pro, MDA, $H_2O_2$, POD and SOD content. Duncan' multiple range tests was used to analyze the data. The lower case letters in the figure represent the comparison at the level of 0.05. The same letter indicates that the difference is not significant, and the different letters indicate that the difference is significant.

during the development of 1258A pollen, but increased significantly in 1258B. However, the content of MDA in P5A was significantly lower than that in maintainer line at SB and MB stages (Fig 3E). This may be due to the early abortion stage of P5A pollen, so the content of MDA is relatively stable in the later stage of pollen development.

The change trend of $H_2O_2$ content was very similar to that of Pro content. In both SB and MB stages, the $H_2O_2$ content of 1258A and P5A were lower than the maintainer line, the $H_2O_2$ content of 1258B and P5B was the highest in MB stage (Fig 3F). The POD content of 1258A and P5A in SB stage was significantly lower than that in maintainer line, and increased significantly in MB stage after pollen abortion, while the POD content in maintainer line did decrease significantly in MB stage (Fig 3G). The SOD content in the anther SB stage of the two male sterile lines was significantly lower than that of the maintainer line. Compared with the SB stage, the SOD content of the maintainer line in the MB stage decreased significantly, but there was no significant change in the anther of the male sterile line (Fig 3H). We noticed that the energy-related substances such as sucrose, soluble sugar and ATP in the anther of the male sterile line were significantly lower than those of the maintainer line, which was obvious in the period before pollen abortion. The contents of antioxidant enzymes such as POD and SOD in the early stage of pollen abortion were lower than those of maintainer lines, which may reduce the ability of male sterile lines to scavenge harmful oxygen free radicals in cells. However, it was found that only 1258A accumulated a large amount of MDA in the early stage of pollen abortion, while the contents of $H_2O_2$ did not accumulate in the two male sterile lines.

## Identification of LncRNA during anther development

The buds of different sizes of *Nsa* CMS (1258A), *Pol* CMS (P5A) and restorer line (P26R) were sequenced with three biological repeats of high-throughput techniques. The sampling criteria were as follows: small flower buds (SB, ≤ 2mm), an early pollen mother cell that corresponded to pollen development, medium-sized flower buds (MB, 4mm), and mature pollen grains that corresponded to pollen development. A total of 1,571,381,820 clean reads were obtained by sequencing all 18 libraries, and the filtered reads accounted for > 98% of the total reads of the

**Table 1. Statistical analysis of RNA sequencing data.**

| Sample | Raw reads number | Clean reads number | clean rate (%) | Mapped reads | Uniquely mapped reads | Multiple mapped reads |
|---|---|---|---|---|---|---|
| 1258_MB1 | 91,660,092 | 90,852,122 | 98.68 | 84,512,689(93.02%) | 48,623,104(57.53%) | 35,889,585(42.47%) |
| 1258_MB2 | 93,758,630 | 92,375,408 | 98.17 | 85,812,889(92.90%) | 51,975,792(60.57%) | 33,837,097(39.43%) |
| 1258_MB3 | 94,158,794 | 93,277,028 | 98.86 | 86,924,096(93.19%) | 52,549,140(60.45%) | 34,374,956(39.55%) |
| 1258_SB1 | 94,725,968 | 93,919,196 | 98.98 | 87,263,728(92.91%) | 58,063,840(66.54%) | 29,199,888(33.46%) |
| 1258_SB2 | 81,061,698 | 80,247,228 | 98.6 | 72,874,840(90.81%) | 48,420,379(66.44%) | 24,454,461(33.56%) |
| 1258_SB3 | 89,211,794 | 88,302,878 | 98.77 | 81,426,030(92.21%) | 55,287,182(67.90%) | 26,138,848(32.10%) |
| P5A_MB1 | 87,332,036 | 86,153,082 | 98.45 | 79,138,646(91.86%) | 54,305,200(68.62%) | 24,833,446(31.38%) |
| P5A_MB2 | 93,114,336 | 91,940,494 | 98.34 | 84,517,750(91.93%) | 57,021,513(67.47%) | 27,496,237(32.53%) |
| P5A_MB3 | 84,745,584 | 83,684,358 | 98.51 | 76,414,029(91.31%) | 51,740,122(67.71%) | 24,673,907(32.29%) |
| P5A_SB1 | 94,973,812 | 93,861,006 | 98.44 | 86,048,676(91.68%) | 56,957,558(66.19%) | 29,091,118(33.81%) |
| P5A_SB2 | 79,267,094 | 78,306,186 | 98.46 | 71,833,159(91.73%) | 47,141,933(65.63%) | 24,691,226(34.37%) |
| P5A_SB3 | 95,425,468 | 94,360,252 | 98.71 | 86,212,339(91.37%) | 59,460,380(68.97%) | 26,751,959(31.03%) |
| P26R_MB1 | 80,782,448 | 79,609,292 | 98.15 | 73,532,658(92.37%) | 46,134,319(62.74%) | 27,398,339(37.26%) |
| P26R_MB2 | 89,756,550 | 88,698,372 | 98.55 | 81,321,478(91.68%) | 52,951,159(65.11%) | 28,370,319(34.89%) |
| P26R_MB3 | 81,936,062 | 80,792,752 | 98.14 | 73,853,484(91.41%) | 46,774,201(63.33%) | 27,079,283(36.67%) |
| P26R_SB1 | 86,272,052 | 84,929,490 | 98.11 | 77,489,271(91.24%) | 52,015,846(67.13%) | 25,473,425(32.87%) |
| P26R_SB2 | 95,733,062 | 94,696,382 | 98.74 | 85,812,147(90.62%) | 56,678,251(66.05%) | 29,133,896(33.95%) |
| P26R_SB3 | 76,377,380 | 75,376,294 | 98.41 | 68,067,660(90.30%) | 45,618,371(67.02%) | 22,449,289(32.98%) |

Clean Reads: reads obtained after quality control filtering; Mapped reads: matching to genome and percentage. Uniquely mapped reads: matching to a unique position in the genome and percentage. Multiple mapped reads: matching to multiple positions in the genome and percentage.

original data, exceeding the 90% clean reads alignment with the reference genome (Table 1). According to the length of transcripts and the value of Fragments Per Kilobases of transcript per Million mapped reads (FPKM), the potential assembled transcripts with lengths < 200bp were filtered. For the assembled transcripts with only one exon, the transcripts with FPKM ≥ 2 were retained, and those with FPKM ≥ 0.5 were retained for multiple exons. The unknown transcript assembled by StringTie software was used for further coding capability filtering, and CPC, CNCI and HMMER software were selected to predict the coding ability. Those with no coding abilities predicted by the three software are novel lncRNAs. In this study, 39,681 novel lncRNAs were identified, and 127,210 known mRNAs and 17,401 known lncRNAs were identified (Fig 4). The sequence information of lncRNAs and mRNAs is shown in S1 Table.

## Differentially expressed lncRNAs and mRNAs in 1258A, P5A, and P26R

The difference in expression of lncRNAs between male sterile and fertile buds before and after abortion (SB) and after abortion (MB) was analysed based on the id of known lncRNAs and novel lncRNAs. The differences in the total RNA were statistically analysed, and the different analytical results of lncRNA were obtained. There were 30,443 lncRNAs differentially expressed in 1258A_SB_vs_1258A_MB, 27,524 lncRNAs differentially expressed in P5A in the SB and MB stages (P5A_SB_vs_P5A_MB), and 31,941 lncRNAs differentially expressed in P26R_SB_vs_P26R_MB in the SB and MB stages (Fig 5A). Considering that the difference of P26R in fertile buds at SB and MB stages is real, there are 4088 and 3198 lncRNAs unique to the two sterile buds at the SB and MB stages, respectively, and 2424 lncRNAs in total (Fig 5A). Simultaneously, we found that there were 5344 and 3469 mRNAs unique to 1258A and P5A male sterile buds at the SB and MB stages, respectively, and 3157 mRNAs (Fig 5B). In

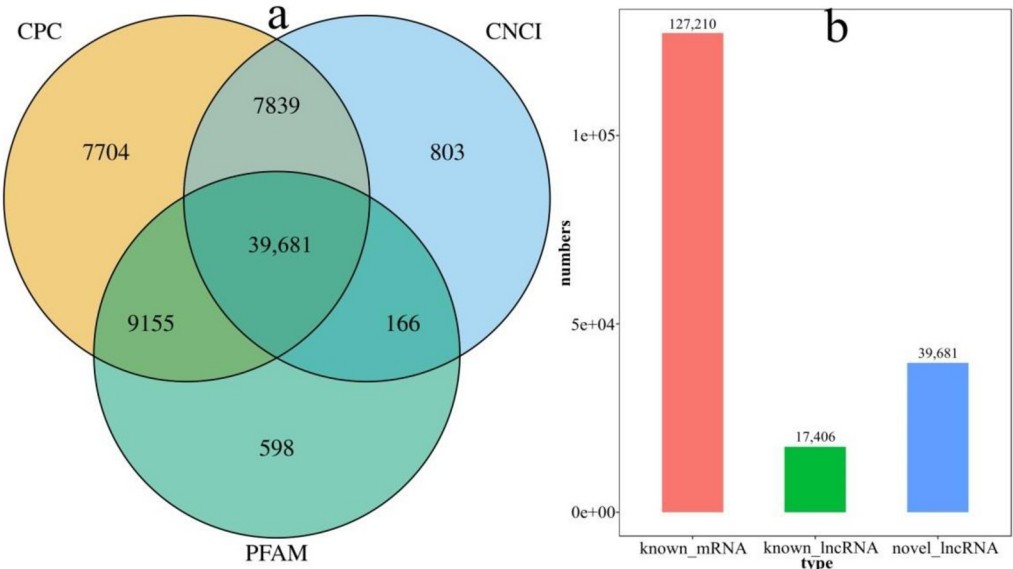

**Fig 4. Number of lncRNA and mRNA.** a: Venn diagram for predicting the number of novel lncRNA by three methods. b: Statistical histogram for transcripts: abscissa is the transcript type; ordinate is the number of transcripts corresponding to that transcript type.

1258A_SB_vs_1258A_MB, there were 2364 up-regulated lncRNAs, 7655 up-regulated mRNAs, 3117 down-regulated lncRNAs and 5526 down-regulated mRNAs (Fig 5C). In P5A_SB_vs_P5A_MB, there were 1479 up-regulated lncRNAs, 3869 up-regulated mRNAs, 1942 down-regulated lncRNAs and 3857 down-regulated mRNAs (Fig 5C). There were 1507 up-regulated lncRNAs, 2925 up-regulated mRNAs, 1002 down-regulated lncRNAs and 2056 down-regulated mRNAs in P26R_SB_vs_ P26R _MB (Fig 5C). The number of down-regulated lncRNAs in the normal fertile buds was less than that of the up-regulated lncRNAs, while the number of down-regulated lncRNAs in the sterile buds was higher than that of the up-regu-lated lncRNAs. The number of lncRNAs and mRNAs of the fertile buds of P26R was higher than that of the down-regulated expression, while 1258A and P5A responded in the opposite manner. Thus, it is apparent that these down-regulated differences in the lncRNAs in male sterile buds are closely related to pollen abortion. The DE lncRNAs and statistical results can be found in S2 Table.

## Analysis of the co-expression of differential lncRNA and differential mRNA during pollen development

The significant correlation of lncRNA-mRNA relationship was screened for functional analy-sisby calculating the correlation between the values of expression of lncRNAs and mRNAs in the stages before and after pollen abortion. A total of 209,979 significantly related lncRNA-mRNA pairs were screened in 1258A_SB_vs_1258A_MB; 69,695 significant lncRNA-mRNA pairs were screened in P5A_SB_vs_P5A_MB, and 19,824 significant lncRNA-mRNA pairs were screened in P26R_SB_vs_P26R_MB (S3 Table). The number of lncRNA-mRNA pairs before and after the abortion of two male sterile buds was significantly higher than that of the fertile buds, which showed that a large number of different lncRNAs had complex biological functions during the process of pollen abortion. Therefore, according to the significant rela-tionship, the corresponding genes of mRNA were analyzed by Gene Ontology (GO) and Kyoto Encyclopedia of Genes and Genomes (KEGG) pathway functional analyses to indirectly

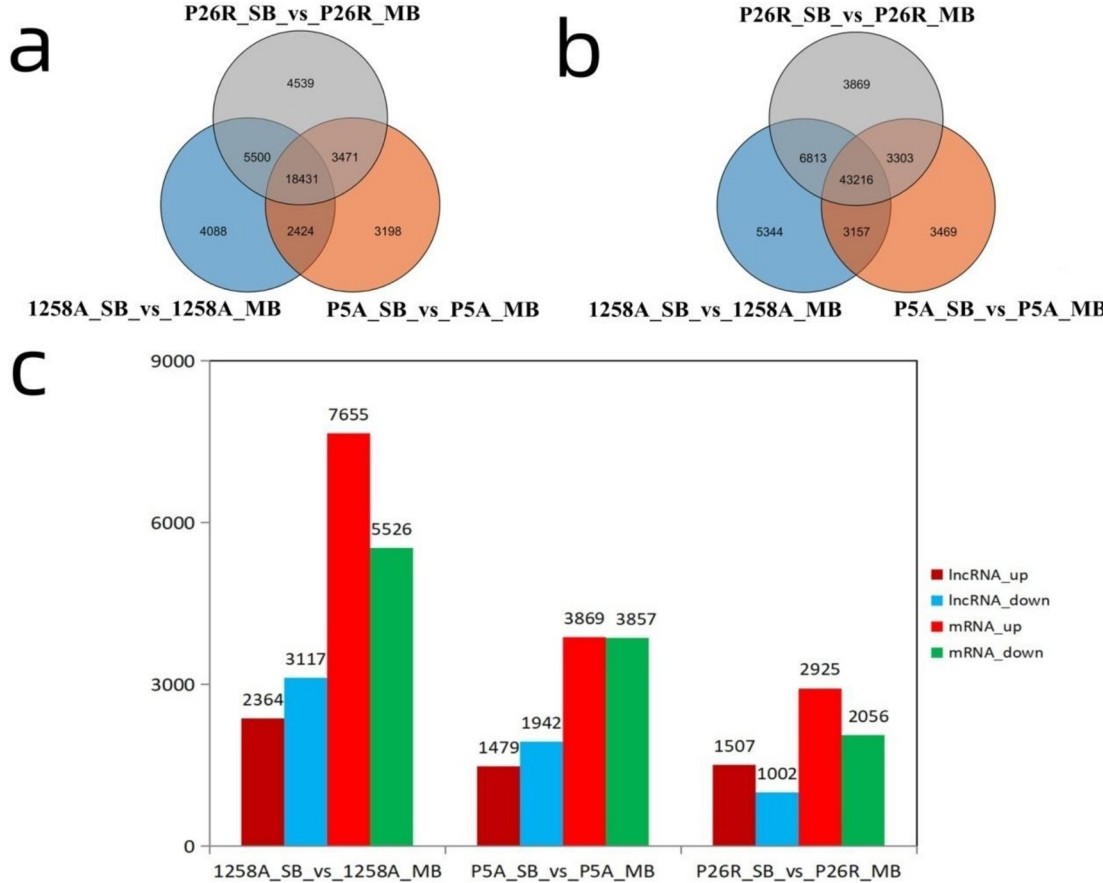

**Fig 5. Statistical analysis of lncRNA and mRNA that were differentially expressed at different stages of pollen development.**
a: Venn diagram of differentially expressed lncRNA in different sizes buds of 1258A, P5A, and P26R. b: Venn diagram of differentially expressed mRNA in differently sized buds of 1258A, P5A, and P26R. c: Statistical results of up-regulated and down-regulated lncRNA and mRNA in different sizes buds of 1258A, P5A, and P26R.

determine the lncRNA function of 1258A and P5A before and after pollen abortion. We focused on the first five GO enrichment results. The results of functional enrichment analysis of differential mRNA gene by GO showed that ribosome biogenesis (GO: 0042254), regulation of cell cycle (GO: 0051726), cell cycle (GO: 0007049), nucleolus (GO: 0005730) and organic hydroxy compound biosynthetic process (GO: 1901617) were enriched by 1258A_SB_vs_1258A_MB (Fig 6A). Acid phosphatase activity (GO: 0003993), lipid localization (GO: 0010876), lipid transport (GO: 0006869), cellular response to phosphate starvation (GO: 0016036) and lipid particle (GO: 0005811) were enriched by P5A_SB_vs_P5A_MB (Fig 6B). Catalytic activity (GO: 0003824), fatty acid metabolic process (GO: 0006631), lipid metabolic process (GO: 0006629), monocarboxylic acid biosynthetic process (GO: 0072330) and monocarboxylic acid metabolic process (GO: 0032787) were enriched by P26R_SB_vs_P26R_MB (Fig 6C). This analysis indicated that the most significant GO subtaxonomy before and after 1258A and P5A pollen abortion and P26R pollen development were completely different.

In addition, based on the KEGG analysis, the genes that corresponded to mRNA in these lncRNA-mRNA pairs were significantly enriched in 30 pathways before and after 1258A and P5A pollen abortion and P26R pollen development, respectively (Fig 7). We found that among the significantly enriched terms, both the two sterile buds and the fertile buds had the largest

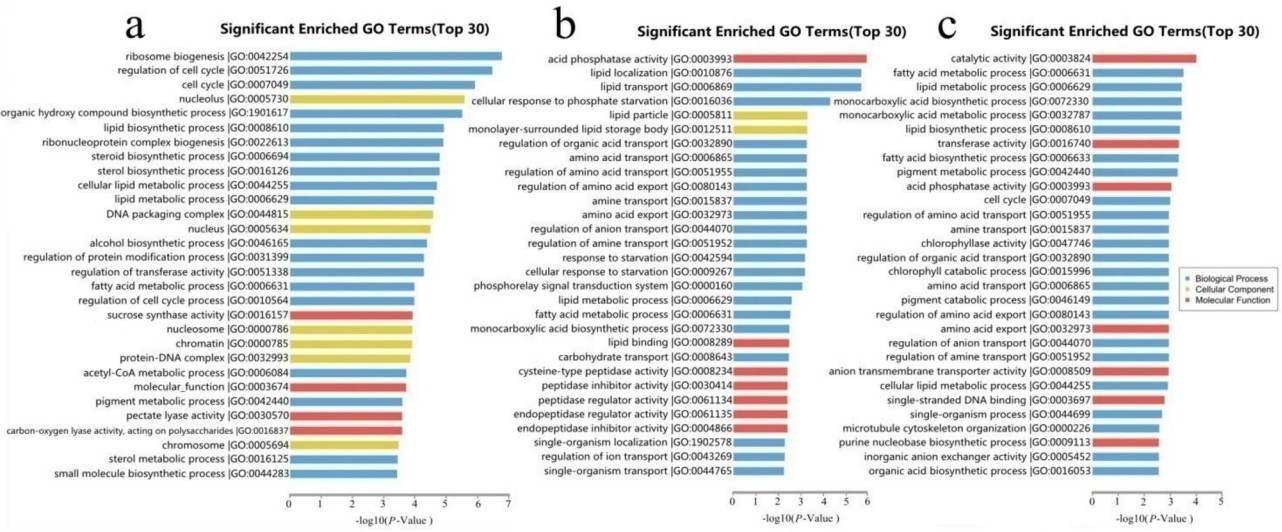

**Fig 6. Gene ontology (GO) function enrichment analysis, GO terms of differentially expressed genes in rapeseed buds.** a: 1258A_SB vs_1258A_MB top 30 GO terms. b: P5A_SB_vs_P5A_MB top 30 GO terms. c: P26R_SB_vs_P26R_MB top 30 GO terms. GO enrichment analysis of DE genes with a *p*-value < 0.05.

number of differential genes in Metabolic pathways. Among the first five pathways of significant enrichment, Biosynthesis of secondary metabolites, Metabolic pathways, Plant hormone signal transduction and alpha-Linolenic acid metabolism were found to be common in 1258A_SB_vs_1258A_MB, P5A_SB_vs_P5A_MB and P26R_SB_vs_P26R_MB. It is hypothesised that these four common pathways could be the real difference between the SB and MB stages of flower buds, which was not the focus of our research. Interestingly, we found that Starch and sucrose metabolism and Sulfur metabolism were significantly enriched in 1258A_SB_vs_1258A_MB and P5A_SB_vs_P5A_MB, and the color was red but not in P26R_SB_vs_P26R_MB. Most of the DE genes in Starch and sucrose metabolism and Sulfur

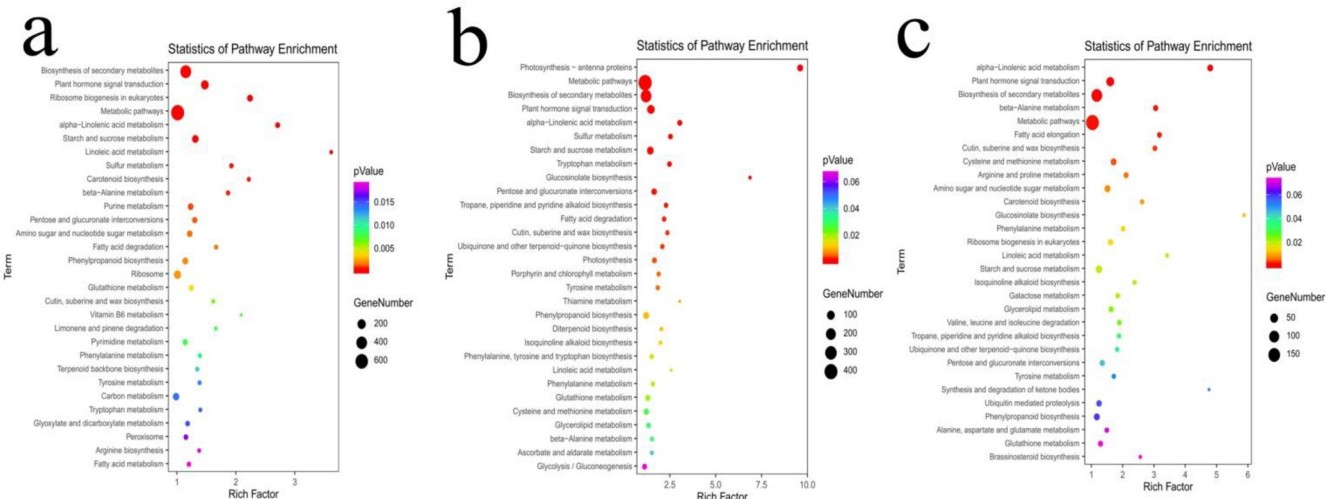

**Fig 7. The top 30 KEGG pathway enrichments of DE genes in rapeseed buds.** a-c: 1258A_SB vs_1258A_MB, P5A_SB_vs_P5A_MB and P26R_SB_vs_P26R_MB KEGG pathway enrichment. The size of each circle represents the number of significant DE genes enriched in the corresponding pathway. A pathway with a *p*-value < 0.05 is considered to be significantly enriched.

metabolism of male sterile buds were down-regulated (S1 Fig in S1 File), so we suspect that these down-regulated genes are probably related to pollen abortion and regulated by some lncRNAs. The KEGG pathway analytical results of 1258A_SB_vs_1258A_MB, P5A_SB_vs_P5A_MB and P26R_SB_vs_P26R_MB are shown in S4 Table.

### Identification and functional analyses of *cis*-target genes of lncRNAs

We analysed the potential targets of lncRNAs in *cis* regulation based on the positional relationship between the lncRNAs and coding genes. There were 277 lncRNAs in 1258A_SB_vs_1258A_MB that had *cis*-targeted regulation, and 240 *cis*-target genes of lncRNAs have been identified. *Cis*-targeting regulation was found in 116 lncRNAs of P5A_SB_vs_P5A_MB, and 101 *cis*-target genes of lncRNAs were identified. *Cis*-targeted regulation was found in 33 lncRNAs of P26R_SB_vs_P26R_MB, and 32 *cis*-target genes of lncRNAs were identified (S5 Table). The results of a functional enrichment analysis of the lncRNAs *cis*-target genes by GO showed that there were substantial differences in significantly enriched GO terms between the male sterile buds and fertile buds before and after pollen abortion, and the highly significantly enriched GO terms were completely different (S6 Table). In the KEGG pathway, the *cis*-target genes of these lncRNAs were enriched the most in Metabolic pathways and Biosynthesis of secondary metabolites (Fig 8). We found that 11 KEGG pathways were enriched in the fertile buds of P26R_SB_vs_P26R_MB, including Taurine and hypotaurine metabolism, Butanoate metabolism, and Ascorbate and aldarate metabolism among others. The male sterile buds of P5A_SB_vs_P5A_MB were enriched in 27 KEGG pathways, including Linoleic acid metabolism and Vitamin B6 metabolism among others. 1258A_SB_vs_1258A_MB was enriched 59 KEGG pathways, which were the most abundant. The important KEGG pathways included alpha-Linolenic acid metabolism, Pyrimidine metabolism, Purine metabolism, Ribosome biogenesis in eukaryotes, Glutathione metabolism, beta-Alanine metabolism and Fatty acid biosynthesis. The results of the KEGG pathway analysis of *cis*-target genes of lncRNAs are shown in S7 Table.

We constructed a Venn diagram of the *cis* target genes between the sterile buds and fertile buds and found that there were five target genes in the pollen abortion process of the two male

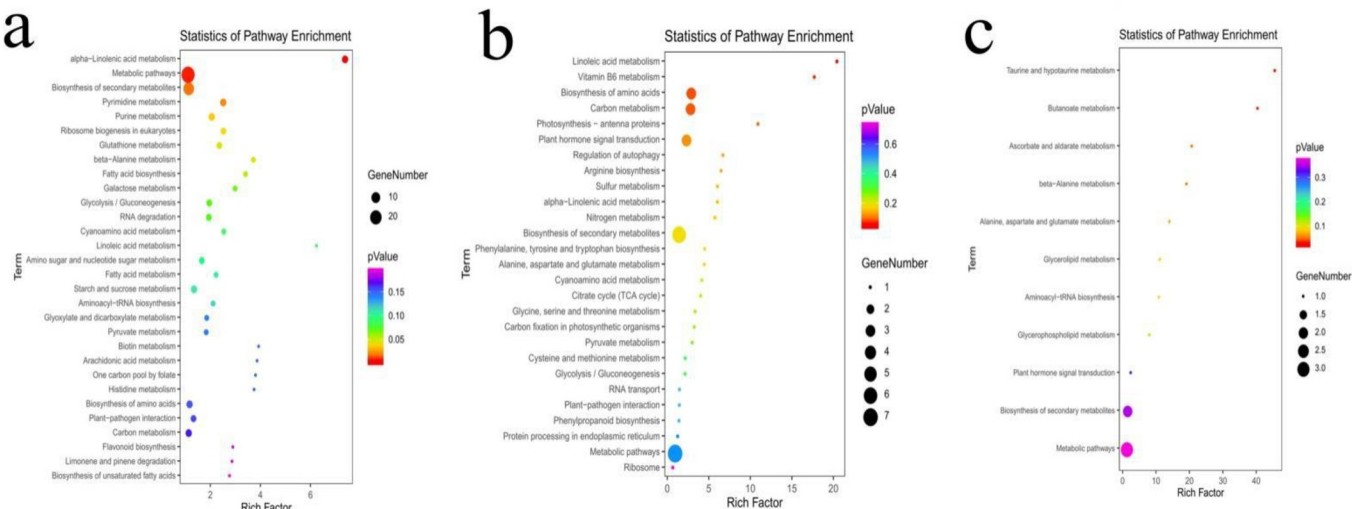

**Fig 8. KEGG pathway enrichments of the *cis*-target genes in rapeseed buds.** a-c: 1258A_SB vs_1258A_MB, P5A_SB_vs_P5A_MB and P26R_SB_vs_P26R_MB KEGG pathway enrichment. The size of each circle represents the number of significant differentially expressed *cis*-target genes enriched in the corresponding pathway. A pathway with a *p*-value < 0.05 is considered to be significantly enriched.

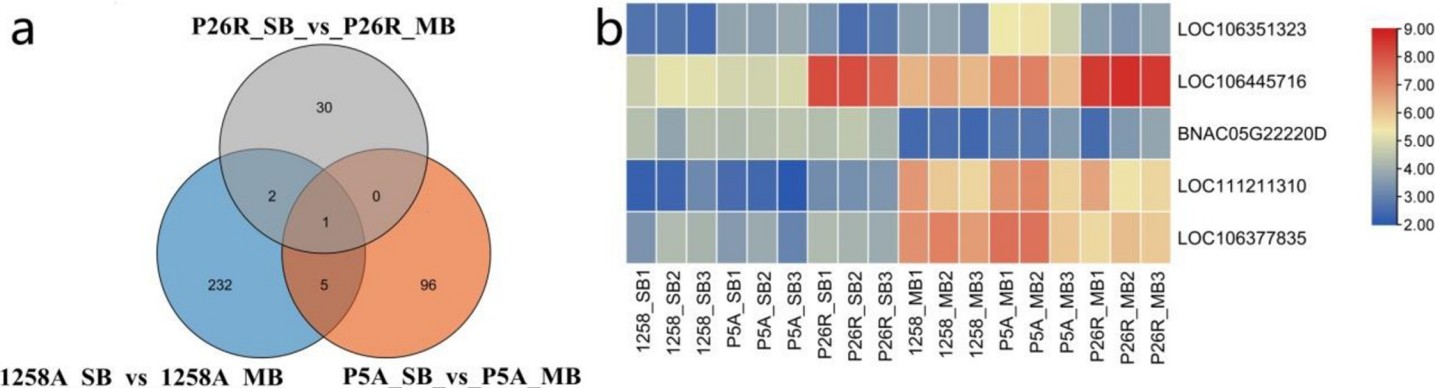

**Fig 9. Venn diagram of the *Cis*-target genes of lncRNAs and the levels of expression of its common target genes in rapeseed buds.**

sterile buds, namely *LOC106351323*, *LOC106445716*, *BNAC05G22220D*, *LOC111211310* and *LOC106377835* (Fig 9A). The *LOC106351323* gene was functionally annotated as ethylene-responsive transcription factor *WIN1*, which was highly expressed in the MB stage of P5A (Fig 9B). The function of *LOC106445716* gene was annotated as β-D-glucopyranosyl abscisate β-glucosidase. The gene was highly expressed in P26R bud during pollen development, and its expression was significantly lower than that of the fertile flower buds during sterile pollen development (Fig 9B). The *BNAC05G22220D* gene encodes the Alba DNA/RNA-binding protein, and it was poorly expressed during pollen development (Fig 9B). The *LOC111211310* and *LOC106377835* genes were functionally annotated as lipoxygenase 2. The levels of expression of these two genes in male sterile buds were similar to those of the fertile buds, and both were up-regulated at the MB stage (Fig 9B). There was no targeted relationship between the five lncRNAs *cis*-target genes in fertile flower buds, so we hypothesized that these five target genes play a role in the pollen abortion of male sterile buds and are regulated by lncRNAs. In particular, the *LOC106445716* gene, as the target gene of *MERGE.18561.2*, was down-regulated during thepollen development of male sterile flower buds, which is closely related to pollen abortion.

## Discussion

### Physiological basis of male sterile in Nsa CMS and Pol CMS

The decrease of energy supply, the decrease of the activity of antioxidant enzymes and the intensification of oxidative stress in cells are common conditions in the process of plant pollen abortion. *Pol* CMS has been proved to be due to energy deficiency caused by mitochondrial gene *orf224*, which suppresses a series of genes that regulate pollen development through nuclear-mitochondrial interaction, leading to the failure of sporogenic cell differentiation [32]. We also detected that sucrose and ATP contents of anthers in *Pol* CMS were significantly lower than those in maintainer lines at the early stage of pollen abortion, which was accompanied by the whole process of pollen development. It confirmed that the energy supply in anthers was insufficient. The energy supply of *Nsa* CMS anther was similar to that of *Pol* CMS, but the soluble sugar content of *Nsa* CMS and *Pol* CMS was not significantly lack at the earlier stage of pollen development. This should be due to the fact that sucrose is the main carbohydrate needed during anther development [46]. Proline is a kind of amino acid with many functions, and it is a storage form of amino acids in flowering plant pollen. In the process of pollen development, proline cooperates with carbohydrates and other substances with high content. It mainly plays the role of providing nutrients, promoting pollen growth and development,

germination and pollen tube extension. The Pro content of *Nsa* CMS and *Pol* CMS during pollen development was significantly lower than that of maintainer lines, which indicated that proline synthesis was blocked during pollen development and could not provide necessary amino acids for pollen development. Insufficient supply of carbohydrates and nitrogen sources is an important reason for pollen abortion of *Nsa* CMS and *Pol* CMS.

MDA is a widely used marker of membrane lipid peroxidation. Its accumulation is related to the biotic and abiotic stresses suffered by organisms, and its content can reflect the level of ROS in plants. Mitochondria are the main producing sites of ROS, and the outbreak of ROS during pollen development of CMS is closely related to pollen abortion [47, 48]. During the whole pollen development of CMS, the MDA content of the male sterile line increased significantly, and the MDA content of the male sterile line was always higher than that of the maintainer line with the excessive accumulation of reactive oxygen species [49–52]. However, it was found that the content of MDA in *Pol* CMS was significantly lower than that in maintainer line before and after pollen abortion, and only *Nsa* CMS had higher MDA accumulation. This may be related to the premature abortion of *Pol* CMS, which occurs in the stage of sporogonia differentiation and there is no further pollen development, so there is no large amount of MDA accumulation. It was found that the $H_2O_2$ content in fertile anthers was higher than that in sterile anthers, and $H_2O_2$ content accumulated continuously during pollen maturation. We speculate that $H_2O_2$ is not the main peroxide, but the accumulation of other products such as superoxides and hydroxyl radicals may lead to the outbreak of ROS. Antioxidant enzymes such as POD and SOD can maintain the balance of intracellular oxygen free radicals and protect cells from ROS damage. Our results showed that the contents of POD and SOD in the two male sterile lines were significantly lower than those in the maintainer lines, which indicated that the oxygen free radicals and some other metabolites which were not conducive to pollen development could not be eliminated in time. Our results showed that the tapetum development was abnormal during *Nsa* CMS abortion, and the dysfunction of reactive oxygen species and antioxidant defense system was very likely to lead to abnormal tapetum degradation and pollen abortion. In this study, it was found that the contents of sucrose and ATP in anthers in male sterile lines were significantly lower than those in maintainer lines.

## Identification and functional verification of lncRNAs during the process of pollen abortion in rapeseed

It has been proven that lncRNAs play an important role in plant growth and development, and their molecular mechanism and biological role in the regulation of pollen development are gradually being revealed. Plant male sterile materials owing to pollen abortion provide a good system to study the expression of genes during pollen development. For example, the differential lncRNAs of pollen abortion was identified in cotton CMS, and the abnormal expression of differential lncRNAs *cis*-target gene was closely related to pollen abortion [24, 25]. The construction of an lncRNA regulatory network further explained that lncRNA could participate in CMS through cellular hormone metabolism and redox reactions. In addition, it was also found that lncRNAs could play a role as a precursor of miRNA and miRNA eTM (endogenous target mimic) in cotton, and *ghr-miR156* could be a regulator of Gh_A08G0866 and Gh_D08G2688 by participating in the development of anther cell walls [53]. In the nuclear male sterile lines of rapeseed, the gene modules and key lncRNAs that regulate reproductive development were identified, and these lncRNAs were highly co-expressed with the genes encoded by their target proteins. It was found that 14 of these lncRNAs were highly co-expressed with known pollen-related coding genes; 15 lncRNAs were predicted to be endogenous targeting simulations of 13 miRNAs, and two lncRNAs were proven to be functional targeting mimics of *miR160* that are

involved in pollen formation and male fertility [31]. *Pol* CMS and *Ogu* CMS are the most widely used CMS systems in rapeseed [9]. Studies on genic male sterile lines and *Pol* CMS and *Ogu* CMS showed that the candidate genes and lncRNAs and their regulatory networks related to pollen abortion differed. For example, most tapetum development-related genes were down-regulated in *Pol* CMS, but only those genes that play a role in tapetum degradation were down-regulated in genic male sterile and *Ogu* CMS lines. Further comparative study indicated that the two lncRNAs were functional precursors of *bra-miR156HG* and *bra-miR5718HG*, respectively. The overexpression of *bra-miR5718HG* in *B. campestris* could affect fruiting [33].

Using male sterile lines to study the regulatory role of lncRNAs in pollen abortion is increasingly being clarified, but different male sterile types often display varying regulatory mechanisms. LncRNAs have not been identified during pollen development in *Nsa* CMS, and their regulatory role has not been reported. Therefore, we selected different sizes of flower buds during the pollen development of *Nsa* CMS and *Pol* CMS and used high-throughput sequencing technology to analyze the whole genome of lncRNAs. A total of 30,443 DE lncRNAs and 4088 unique lncRNAs were identified during 1258A (*Nsa* CMS) pollen abortion. A total of 27,524 DE lncRNAs and 3198 unique lncRNAs were identified during P5A (*Pol* CMS) pollen abortion. There were 2424 lncRNAs in 1258A and P5A. Compared with fertile buds, the number of down-regulated lncRNAs in the sterile buds was higher than that of the up-regulated lncRNAs. These down-regulated lncRNAs are likely to regulate pollen abortion. For example, lncRNA *zm401* is the primary regulator of the expression of key genes in pollen development, and transgenic plants with down-regulated *zm401* expression display atypical tapetum and microspore formation, which leads to the production of sterile pollen [54, 55].

## Interaction between LncRNA and mRNA during pollen abortion in rapeseed

LncRNA has important biological significance. LncRNAs can regulate gene expression and comprise one of the molecular mechanisms of post-transcriptional regulation and regulation of protein function. LncRNAs can promote gene expression by competing with miRNAs for specific binding sites in the non-coding region of mRNAs and preventing transcriptional inhibition caused by miRNAs [17, 56, 57]. In this study, we found that the mRNAs comprise a rich transcript. There were 5344 and 3469 mRNAs unique to 1258A and P5A male sterile buds at the SB and MB stages, respectively, and 3157 mRNAs. In fertile buds, the number of lncRNAs and mRNAs of P26R was higher than that of down-regulated expression. The number of lncRNAs and mRNAs of 1258A and P5A in the male sterile buds was higher than that of the up-regulated expression. It was found that there were LncRNA-mRNA interactions in rapeseed pollen wall construction, tapetum development, anther cell differentiation and pollen meiosis [33]. The interaction between mRNAs and ncRNAs is common during anther and pollen development. An ncRNAs analysis of the male sterile system in cotton and rapeseed showed that most of the expressed genes and ncRNAs played an active role in pollen development, and their low expression led to pollen abortion [25, 31, 33, 53, 58]. We screened 209,979 significant lncRNA-mRNA pairs during the SB and MB stages of 1258A pollen development, 69,695 significant lncRNA-mRNA pairs in P5A and 19,824 significant lncRNA-mRNA pairs in P26R. The number of lncRNA-mRNA relationships between the two male sterile pollen development were significantly higher than that of P26R fertile buds, so we hypothesized that there were many differences before and after the abortion of male sterile buds. LncRNAs play an important role during the process of pollen abortion. Therefore, according to the significant relationship, the corresponding genes of mRNA were analyzed by GO and pathway functional analyses to determine the role of lncRNAs in 1258A and P5A pollen abortion. We found that

the most significant GO subtaxonomy during the pollen development of 1258A and P5A before and after pollen abortion and P26R were completely different. This GO difference is consistent with the results of previous transcriptome analyses in *Pol* CMS and *Nsa* CMS [1, 8, 9, 32, 35]. However, we also found that the two male sterile types had something in common. A KEGG analysis of different flower buds of *Pol* CMS and *Nsa* CMS showed that both Starch and sucrose metabolism and Sulfur metabolism were significantly enriched in the KEGG pathway, and most of the differentially expressed genes were down-regulated. Therefore, we hypothesised that the down-regulation of these differential genes are closely related to affect the normal development of pollen and could be regulated by some lncRNAs.

## Abnormal metabolism of Starch and sucrose could be an important factor that leads to pollen abortion

Plants produce carbohydrates through photosynthesis, which provide the necessary energy supply for the development of pollen. Starch is an important energy storage material in plants, and sucrose is the primary source of energy during the process of anther development [46]. Studies have shown that an insufficient supply of sugar can lead to pollen abortion, and sugar metabolism affects pollen development in plants [59, 60]. In this study, we found that sucrose and ATP contents in anthers were significantly lower in sterile lines than in maintainers. A KEGG analysis indicated that several differential genes of *Pol* CMS and *Nsa* CMS Starch and sucrose metabolism were down-regulated. The down-regulation of these differential genes could lead to an insufficient supply of energy during pollen development, which, in turn, leads to male sterility. The results showed that the energy content in flower buds of male sterile lines was generally lower than that of the maintainer lines, and the contents of sucrose, starch and ATP in male sterile buds were significantly lower than those in fertile buds [47, 49, 51, 61]. Increasing amounts of evidence show that an insufficient supply of energy and a decrease in the content of ATP will further aggravate the oxidative stress response of cells, resulting in the excessive accumulation of reactive oxygen species and pollen abortion [49, 51, 61, 62]. Our previous studies showed that the contents of sucrose, soluble sugar and ATP in anthers of male sterile lines were very low, and there were significant differences in sugar metabolism and energy metabolism genes, such as β-glucosidase-related genes [35]. It was also found that starch biosynthesis during pollen maturation in the CMS lines of maize was related to the abnormal expression of a lot of glucose metabolism genes [63]. Therefore, we hypothesized that Starch and sucrose metabolism is closely related to rape pollen development, and its metabolic abnormality could be an important factor in pollen abortion.

The down-regulated expression of β-glucosidase-related genes could affect the supply of energy sources for pollen development. β-glucosidase can hydrolyze starch and cellulose, and the deglycosylation of glycosides leads to the accumulation of glucose, which forms uridine-5-diphosphate glucose (uridine-5'-diphospho-glucose, UDPG), resulting in the accumulation of UDPG [64]. UDPG serves as a glucosyl donor in many enzymatic glycosylation processes, such as oligosaccharides, polysaccharides and glycoproteins [65]. Our previous transcriptome analysis of 1258A and its maintainer lines also showed that the β-glucosidase-related genes were abnormally expressed in sterile buds [35]. Therefore, we concluded that the genes related to β-glucosidase play an important role in the development of pollen in rapeseed. Interestingly, we noticed that a β-glucosidase gene, *LOC106445716*, is the target gene of the differential lncRNA between 1258A and P5A, and there is *cis*-targeted regulation between this gene and *MERGE.18561.2*. Compared with fertile buds, the level of expression of *LOC106445716* in two sterile buds was significantly down-regulated during pollen development, which could be an important factor that leads to pollen abortion.

## Conclusions

In this study, we made a detailed comparative study on the characteristics of pollen abortion between *Nsa* CMS (1258A) and *Pol* CMS (P5A) by cytological observation and physiological and biochemical indexes. These two male sterile lines are faced with serious lack of energy supply during pollen development, while *Nsa* CMS has a large amount of ROS accumulation during pollen development. The in-depth study of lncRNAs in two male sterile lines showed that there were 30,443, 27,524 and 31,941 lncRNAs differentially expressed in 1258A, P5A and P26R at SB and MB stages, respectively. The specific lncRNAs of the two male sterile buds at the SB and MB stages were 4088 and 3198, respectively, and the total lncRNAs were 2424. We screened out significantly correlated lncRNA-mRNA relationship pairs for functional analysis and found that Starch and sucrose metabolism and Sulfur metabolism were significantly enriched in 1258A and P5A pollen abortion but not significantly enriched in the fertile bud SB and MB stages. Most of the DEGs in these two pathways were down-regulated. We performed Gene ontology and KEGG enrichment analyses on the potential *cis*-target genes of these differential lncRNAs, which showed that they played an important role in the regulation of anther development in rapeseed. We identified five common lncRNAs *cis*-target genes in the CMS system of two male sterile types, namely *LOC106351323*, *BNAC05G22220D*, *LOC106445716*, *LOC111211310* and *LOC106377835*. Among them, the β-glucosidase-related gene *LOC106445716*, as the target gene of *MERGE.18561.2*, was down-regulated in the pollen development of male sterile flower buds, which is closely related to pollen abortion. In this study, by comparing the characteristics of 1258A and P5A pollen abortion, combined with the functional analysis of lncRNAs in the process of pollen abortion, it provides a new perspective for lncRNAs to participate in the regulation of *Nsa* CMS and *Pol* CMS pollen abortion.

## Supporting information

**S1 Table.**
(XLSX)

**S2 Table.**
(XLSX)

**S3 Table.**
(XLSX)

**S4 Table.**
(XLSX)

**S5 Table.**
(XLSX)

**S6 Table.**
(XLSX)

**S7 Table.**
(XLSX)

**S1 File.**
(DOCX)

## Author Contributions

**Data curation:** Man Xing, Zechuan Peng, Mei Guan.

**Formal analysis:** Man Xing, Mei Guan.

**Funding acquisition:** Chunyun Guan.

**Investigation:** Zechuan Peng.

**Methodology:** Chunyun Guan, Mei Guan.

**Project administration:** Mei Guan.

**Supervision:** Chunyun Guan, Mei Guan.

**Writing – original draft:** Man Xing.

**Writing – review & editing:** Man Xing, Chunyun Guan, Mei Guan.

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
