## [Decision Letter · Decision Letter 0]

26 Jan 2023

PONE-D-22-30876Comparative Study on Abortion Characteristics of Nsa CMS and Pol CMS and Analysis of Long non-coding RNAs related to Pollen abortion in Brassica napusPLOS ONE

Dear Dr. Guan,

Thank you for submitting your manuscript to PLOS ONE. After careful consideration, we feel that it has merit but does not fully meet PLOS ONE’s publication criteria as it currently stands. Therefore, we invite you to submit a revised version of the manuscript that addresses the points raised during the review process.

We look forward to receiving your revised manuscript.

Kind regards,

Maoteng Li

Academic Editor

PLOS ONE

Journal Requirements:

Reviewers' comments:

Reviewer's Responses to Questions

**Comments to the Author**

1. Is the manuscript technically sound, and do the data support the conclusions?

Reviewer #1: Yes

Reviewer #2: Yes

2. Has the statistical analysis been performed appropriately and rigorously? 

Reviewer #1: Yes

Reviewer #2: Yes

3. Have the authors made all data underlying the findings in their manuscript fully available?

Reviewer #1: Yes

Reviewer #2: Yes

4. Is the manuscript presented in an intelligible fashion and written in standard English?

Reviewer #1: Yes

Reviewer #2: Yes

5. Review Comments to the Author

Reviewer #1: Cytoplasmic male sterile system(CMS) in rapeseed is vital methods for the utilization of heterosis in the breeding practices. This paper identified long non-coding RNAs in two widely applied types of CMS, Nsa CMS and Pol CMS, via high-throughput sequencing. This work was significantly research and fulfill the gap of lncRNAs participating in the regulation of Nsa CMS and Pol CMS pollen abortion. However, there are some problems, which must be solved before it is considered for publication.

1.Firstly, your manuscript needs careful editing and particular attention to English grammar, spelling, and sentence structure, such as:

2.In “plant materials” section, there occurs inconsistent font format in this paragraph, please also check all the paper such as the title of “Transcriptome Sequencing”, “Real-time quantitative reverse transcription PCR (qRT-PCR) Validation” in the method.

3.In “observation of floral organ and anther development” space should be added after comma in the on the fourth line. Also check all of your manuscript.

4.Also some sentences are confusing in the manuscript, the authors should change it, for example “There was only pollen in 1258B anther, but no pollen in 1258A.” Do you mean 1258B can produce pollen but 1258A can’t.

5.The abstract was redundancy and the author could summary your results in short.

6.In the method section, “Determination of physiological indexes in anthers” the numbers of biological replicate was unknown and the method of an analysis was not mentioned yet. For example, in Figure 3, which method was used, t-test or others? Also check it in other part.

7.In Figure8, “KEGG pathway” in a, b and c can be removed.

Reviewer #2: Xing et al., used Nsa CMS (1258A) and Pol CMS (P5A) as materials to do cytological, physiological and biochemical characteristics analysis during pollen abortion, and carried out high-throughput sequencing of flower buds of different sizes before and after pollen abortion. The results provided useful information for CMS research, while there are some points needed to be revised as followings.

1. Method part:

(1) Materials: What’s are P5A and P5B should be added.

(2) Transcriptome Sequencing: how many replicates should be added. And the restore line was used for sequencing in the results part, so this information should be added in this part.

(3) Determination of physiological indexes in anthers: the detailed methods of these physiological and biochemical indexes should be described.

Results part:

(1) The first part “Morphological and cytological characteristics of Nsa CMS and Pol CMS”,

In the first paragraph, the authors could rewrite the content and change the Figure 1A before Figure 1B.

(2) qRT-PCR method was described in the methods part, while there was no corresponding results part. The corresponding result part should be added.

Some minor points:

(1) P8： The sentence A total of 865 DE lncRNAs were identified, the word “DE” should be added with the full name

(2) P10: In the part of “Transcriptome assembly and lncRNA identification”, the first word “Fastp” should add the reference.

6. PLOS authors have the option to publish the peer review history of their article (what does this mean?). If published, this will include your full peer review and any attached files.

Reviewer #1: No

Reviewer #2: No

---

## [Author Response · Author response to Decision Letter 0]

13 Mar 2023

Response to Reviewer1：

Thank you very much for your comments！

1. First, we made a complete revision of the English grammar and structure in the manuscript.

2. We condensed the abstract and briefly summarized the results.

3. We mean that pollen can be produced normally in the anthers of the maintainer line (1258B) and not in the sterile line (1258A).

4. We added the biological repetition of the methods section and the differential analysis method in Figure 3.

5. We removed the "KEGG pathway" section in Figures 7 and Figures 8.

You may review our revised manuscript, thank you very much!

Response to Reviewer2：

Thank you very much for your comments！

1. Method part:

（1）We have added information on plant materials（P5A and P56B） according to your suggestion.

（2）We improved transcriptome sequencing to add biological duplicates. We added the method of determining physiological and biochemical indexes in anthers and improved the method information.

2. Results part:

（1）We rewrote the“Morphological and cytological characteristics of Nsa CMS and Pol CMS” part.The results of qRT-PCR are shown in the attached figure s2.

（2）We added the full name of the word "DE" and the reference to the "Fastp" section.

Thank you very much for your suggestion. We have revised the manuscript for your review.

---

## [Decision Letter · Decision Letter 1]

28 Mar 2023

Comparative Study on Abortion Characteristics of Nsa CMS and Pol CMS and Analysis of Long non-coding RNAs related to Pollen abortion in Brassica napus

PONE-D-22-30876R1

Dear Dr. Guan,

We’re pleased to inform you that your manuscript has been judged scientifically suitable for publication and will be formally accepted for publication once it meets all outstanding technical requirements.

Kind regards,

Maoteng Li

Academic Editor

PLOS ONE

Additional Editor Comments (optional):

Reviewers' comments:

Reviewer's Responses to Questions

**Comments to the Author**

1. If the authors have adequately addressed your comments raised in a previous round of review and you feel that this manuscript is now acceptable for publication, you may indicate that here to bypass the “Comments to the Author” section, enter your conflict of interest statement in the “Confidential to Editor” section, and submit your "Accept" recommendation.

Reviewer #1: All comments have been addressed

Reviewer #2: (No Response)

2. Is the manuscript technically sound, and do the data support the conclusions?

Reviewer #1: Yes

Reviewer #2: Yes

3. Has the statistical analysis been performed appropriately and rigorously? 

Reviewer #1: Yes

Reviewer #2: Yes

4. Have the authors made all data underlying the findings in their manuscript fully available?

Reviewer #1: Yes

Reviewer #2: Yes

5. Is the manuscript presented in an intelligible fashion and written in standard English?

Reviewer #1: Yes

Reviewer #2: (No Response)

6. Review Comments to the Author

Reviewer #1: The authors have revised the manuscript according to my requirements and answered all of the questions, I think this manuscript can be accepted by this journal.

Reviewer #2: The authors revised the ms according to the suggestions, so what my suggestion is it could be accepted in the current form.

7. PLOS authors have the option to publish the peer review history of their article (what does this mean?). If published, this will include your full peer review and any attached files.

Reviewer #1: No

Reviewer #2: No

---

## [Editor Report · Acceptance letter]

3 Apr 2023

PONE-D-22-30876R1 

Comparative Study on Abortion Characteristics of *Nsa* CMS and *Pol* CMS and Analysis of Long non-coding RNAs related to Pollen abortion in *Brassica napus*

Dear Dr. Guan:

I'm pleased to inform you that your manuscript has been deemed suitable for publication in PLOS ONE. Congratulations! Your manuscript is now with our production department. 

Kind regards, 

on behalf of

Dr. Maoteng Li 

Academic Editor

PLOS ONE